**Data Availability Statement:** The interview transcripts used as source of data in this paper represents a minority population, people with

# How do people with disabilities in three regions of Guatemala make healthcare decisions? A qualitative study focusing on use of primary healthcare services

Goli Hashemi[1,2]*, Mary Wickenden[3], Ana Leticia Santos[4], Hannah Kuper[1]

**1** International Centre for Evidence in Disability, London School of Hygiene and Tropical Medicine, London, United Kingdom, **2** Department of Occupational Therapy, Samuel Merritt University, Oakland, California, United States of America, **3** Institute of Development Studies, University of Sussex, Brighton, United Kingdom, **4** Blitz Language, Guatemala City, Guatemala

* Goli.Hashemi@lshtm.ac.uk

## Abstract

Research has demonstrated that persons with disabilities, despite their greater need for healthcare services, often experience greater barriers to accessing healthcare including primary healthcare. Data and interventions on access to primary healthcare for persons with disabilities to date appear to concentrate more on access and quality issues once the person with a disability has initiated the healthcare seeking process, with less emphasis on how healthcare decisions are made at the personal or household level when one identifies a healthcare need. The aim of this study was to investigate how healthcare decisions are made by people with disabilities when they first identify a healthcare need. It is anticipated that gaining a better understanding of how such decisions are made will facilitate the development of interventions and approaches to improve access to primary healthcare services for this group. A qualitative study was undertaken in Guatemala. In-depth interviews were undertaken with twenty-seven adults with disabilities, including men and women with a range of impairment types and ages. Data were analyzed using thematic analysis to identify themes that influence the decision-making processes in accessing primary healthcare services for people with disabilities. Thematic analysis of the interviews along with exploration of three existing access to health frameworks and models, led to identification of four primary themes and development of a new conceptual framework highlighting the complex decision-making process undertaken by people with disabilities about whether to seek primary healthcare services or not when confronted with a healthcare concern. The themes include perceived severity of illness and need for treatment, personal attributes, societal factors, and health system characteristics. Using this new conceptual framework will facilitate the development of more effective policies and interventions to improve access to primary healthcare services for people with disabilities through greater understanding of the complex network of variables and barriers.

disabilities, in areas of Guatemala where they can be easily identifiable based on the details of their characteristics or experiences shared. Given the risk for identification, we would like to request not sharing the interview manuscripts in any data repositories. This is especially important since the participants engaged in the study based on the promise that their personal information and details will not be identifiable or shared and will be kept strictly confidential. However, given your request for an institutional email and access beyond the author, the excerpts from the interviews will be stored at the London School of Hygiene and Tropical Medicine's data repository for access by request only. In order to request access to the data the interested parties can email to request the data.

**Funding:** This work was supported by CBM International (CBM-International to HK)as part of a larger project on access to healthcare for persons with disabilities by the International Center for Evidence on Disability (ICED) at the London School of Hygiene and Tropical Medicine (LSHTM). The funders had no role in study design, data collection and analysis, decision to publish, or preparation of the manuscript.

**Competing interests:** The authors have declared that no competing interests exist.

# Introduction

The 17 Sustainable Development Goals (SDG), adopted by all UN member states in 2015, aim to achieve key milestones by 2030. The SDGs recognize that there is a complex interplay of structural forces that influence an individual's status in society and their wellbeing. Goal three, aiming to 'ensure healthy lives and promotion of well-being for all at all ages', focuses primarily on two aspects of health: reducing mortality and disease and improving access to quality healthcare services for all [1]. This Goal, while extremely important, is an ambitious aspiration as the World Health Organization (WHO) estimates that at least half of the world population or 3.75 billion people worldwide lack access to healthcare services [2].

While there have been some improvements to healthcare access in some parts of the world, globally and nationally progress has been uneven. Access to healthcare is particularly difficult for certain marginalized groups defined by age, gender, income, ethnicity, sexual orientation, and disability. People with disabilities make up about 15%, or one billion, of the world population. The WHO estimates that half of people with disabilities are not able to afford health care, and they are 50% more likely to suffer catastrophic health expenditure due to both the direct and indirect costs of accessing healthcare services. Given these statistics, it will be challenging to meet Goal three of the SDG if people with disabilities continue to be left behind [3].

An increased focus on access to healthcare services for people with disabilities began with the launch of the UN Convention on the Rights of Persons with Disabilities (UNCRPD) in 2006. This treaty reinforces the rights of people with disability, including their right to equal access to health and rehabilitation services [4]. Latterly there have been increasing numbers of publications on this topic [5,6]. Research has demonstrated that people with disabilities often experience greater barriers to accessing healthcare (both general and disability-related), than others in the population, despite their greater need for healthcare services [7–9]. Evidence from High, Middle and Low Income Countries (LMIC) suggests that in comparison to those without disabilities, people with disabilities attend fewer routine health examinations [10], and are less likely to receive preventative care [6,8]. Evidence also suggests that when people with disabilities do seek healthcare, they receive poorer quality services [9,11,12] and incur greater expenses [12]. A recent meta-synthesis of qualitative research on barriers to accessing primary healthcare by people with disabilities in LMICs suggests that the choice to seek healthcare services or not, as well as the quality of intervention provided by primary healthcare providers, are influenced by three types of barriers: cultural beliefs or attitudinal barriers, informational barriers, and practical or logistical barriers [5]. While these categorizations can be valuable, they do not provide us with the details of potential entry-points and interventions to address these barriers.

While various researchers have focused on how healthcare "access" can be broadly conceptualized, none of them have focused on access for particular populations. One example is the Conceptual Framework of Access to Healthcare by Levesque at al. [13], which provides a structure for identification of access and barriers along the healthcare-seeking journey. According to this framework (Fig 1), the journey starts with the identification of a healthcare need, followed by perception of need and desire for care, seeking healthcare services, reaching healthcare resources, obtaining or using healthcare services and actually being offered services appropriate to the care needed [13]. The framework identifies five dimensions of accessibility of services (supply side determinants) and five corresponding dimensions which describe the abilities of consumers to access healthcare services (demand side determinants). Using this framework to match the various barriers commonly experienced by people with disabilities to where they occur in the healthcare seeking process can potentially help focus relevant interventions to improve healthcare access at the appropriate stages. Data from research to date seems

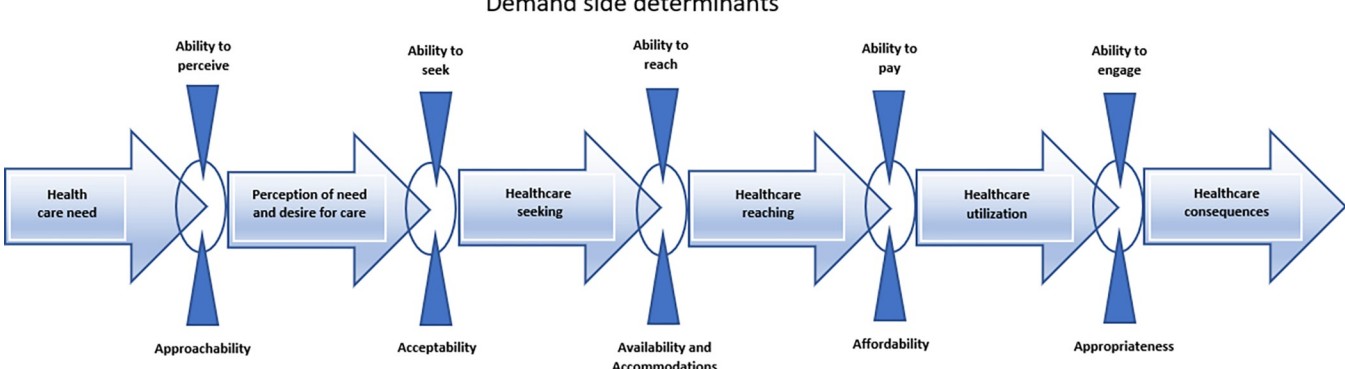

**Fig 1. Framework for access to healthcare adapted [13].**

to concentrate more on access and quality issues once the healthcare seeking process has initiated, with less emphasis on how healthcare decisions are made at the personal or household level [5].

There is a need to further understand the dilemmas people face when deciding whether to and how to seek healthcare services in the first place. Like everyone else, people with disabilities' decisions about accessing healthcare start from within the home, first with the identification of a possible healthcare need, followed by the need to decide whether and how to address this need, as represented by the first two stages in the framework by Levesque et al. [13]. Given their frequently disadvantaged position in society, and related poverty, people with disabilities and their families often need to make tough decisions about where to use their limited resources [3]. This decision-making process might not only result in delays in seeking healthcare services but may also be influenced by prioritizing other family needs or aspects of life over health. It may be deemed that it is not viable or worthwhile to seek primary healthcare services due to the combination of scarcity of resources such as money and time, the anticipated stress of dealing with various logistical factors, such as inaccessible public transport system, the excessive costs involved in arranging alternatives and finally the degree of perceived severity of the problem and expected benefits that may be received after all these costs.

Two other models are also found to be relevant here: Kleinman's Explanatory Model of Illness [14] and the Health Capability Model by Ruger et al. [15]. The Explanatory Model of Illness (Fig 2), developed by Kleinman in the 1970s, uses an anthropological lens to focus on how culture (building on meaning, values, and norms) shapes social reality and personal experiences and thus influences health (and illness) perceptions and behaviors in response to illness [16]. The model emphasizes what Kleinman calls the "popular sector"- consisting of the individual, family, social, and community and how it shapes beliefs, decision making, choices and the relationship and interactions between institutions and settings such as a family and the healthcare sector [16].

The Health Capability Model (Fig 3), developed in 2010, is based on the capability model focusing on an individual's capability to achieve the kind of life they value [15]. The model implies that one's ability to be healthy is based on the dynamic interaction between one's agency and disposition related to facilitators and barriers to accessing health [15]. While the social and structural context still influence one's agency and disposition, the balance between the interactions in this model appears to lack recognition of how meanings and values are developed and the powerful influence they may have on one's agency. This disconnection may be particularly relevant for minority or more marginalized populations, such as people with disabilities. We

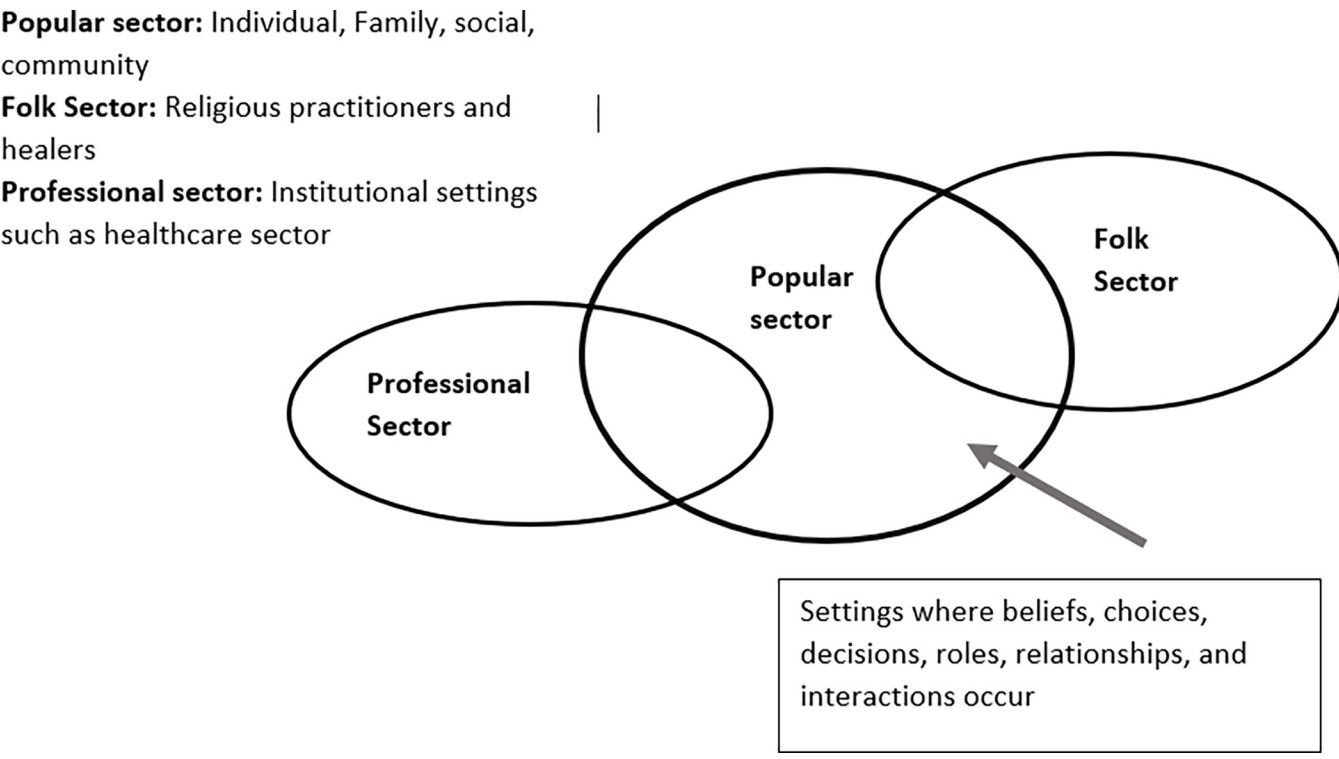

**Popular sector:** Individual, Family, social, community

**Folk Sector:** Religious practitioners and healers

**Professional sector:** Institutional settings such as healthcare sector

Professional Sector

Popular sector

Folk Sector

Settings where beliefs, choices, decisions, roles, relationships, and interactions occur

**Fig 2. Explanatory model of illness adapted [14].**

propose that combining these two models (i.e. incorporating both capability and beliefs) may work well to describe the ability to perceive need and decisions to seek care in the first two stages of the healthcare framework by Levesque et al. [13] for people with disabilities.

The aim of this study is to investigate how healthcare decisions are made by people with disabilities in Guatemala. It is anticipated that gaining a better understanding of how such decisions are made will facilitate the development of interventions and approaches to improve access to primary healthcare services for this group.

## Methods

### Ethics statement

Ethical approval was received from the Ethics Board at the London School of Hygiene and Tropical Medicine in London, England (Reference #: 17627 /RR/14983) and The Institucional de Ética-INCAP (Instituto de Nutrición de Centro América y Panamá) in Guatemala (Reference #: CIE-REV No. 068/2017) To ensure protection of the rights of all participants, prior to the beginning of interviews, an informed consent sheet in Spanish was read and translated item by item to each of the participants. Caregivers were only included in the interview process if the primary participant had consented to the interview and was able to demonstrate capacity that research was being conducted based on their experiences of accessing healthcare services and ascended to the caregiver being part of the interview process.

### Inclusivity in global research

Additional information regarding the ethical, cultural, and scientific considerations specific to inclusivity in global research is included in the Supporting Information (S1 Checklist).

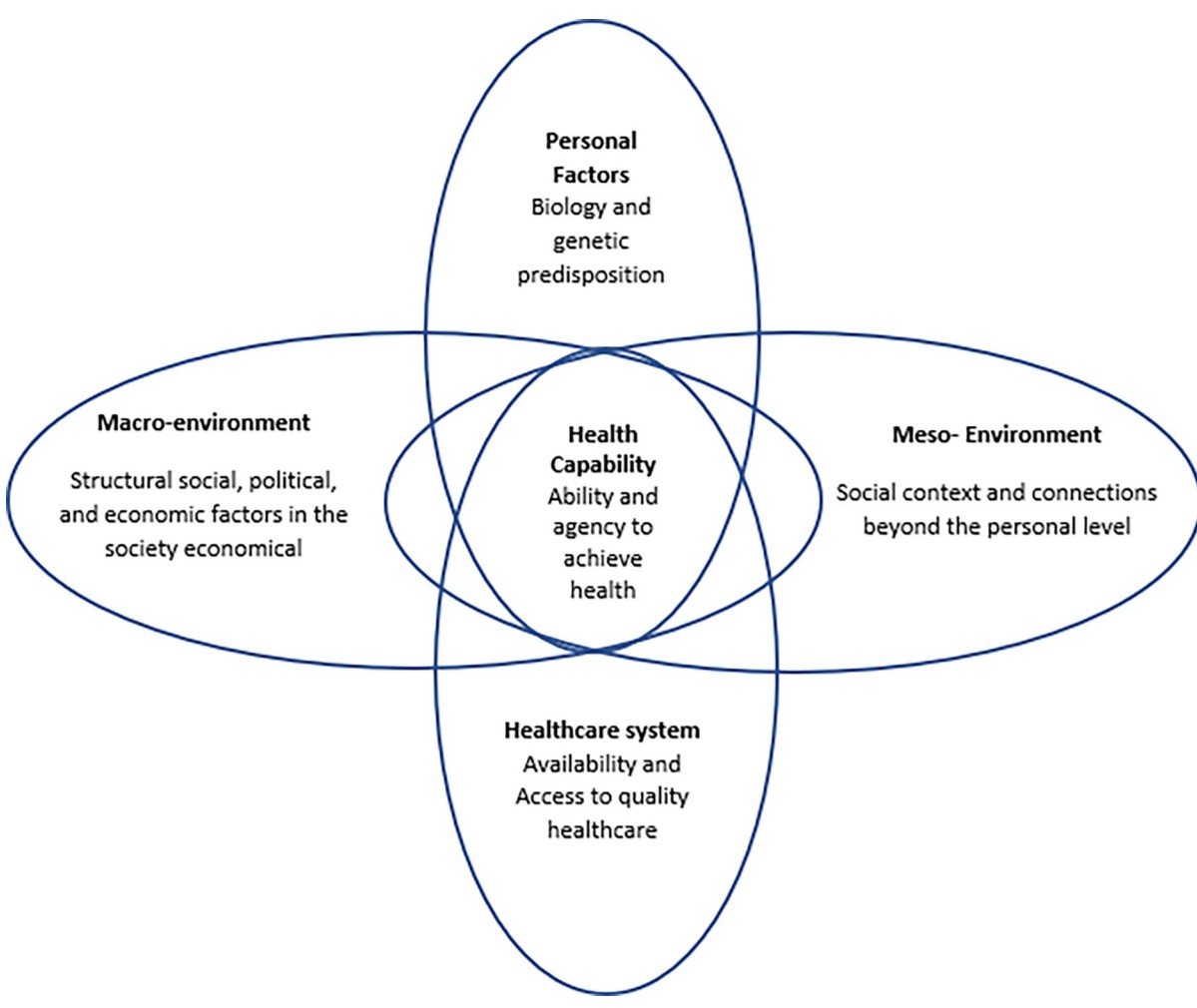

**Fig 3. Health capability model adapted [15].**

## Summary of design

A qualitative study was undertaken in Guatemala during two separate periods in October 2017 and March 2018. In-depth interviews were conducted with a total of twenty-seven adults with disabilities, including men and women with a range of impairment types and ages. Data were analyzed using thematic analysis to identify themes that influence the decision-making processes in accessing primary healthcare services.

## Context

Guatemala has a population of 16.6 million and is administratively divided into 22 departments [17]. It has the largest economy in Central America but is also the fifth poorest country in Latin America and the Caribbean region with some of the highest rates of poverty and inequality [17]. Spanish is recognized as the official language, yet there are 22 ethnic Mayan languages with their own dialects spoken across the country, with most of the indigenous population being monolingual [18]. Guatemala is frequently impacted by natural disasters, ranging from volcanic eruptions, droughts, and storms that result in flooding or mudslides [18]. Although it has a centralized healthcare system, this is very fragmented with funding gaps resulting in inequalities in access to care [18] and increased likelihood of poor care for those in

more rural areas. A national survey estimated that the all age prevalence of disability in 2016 was 10.2% with about 31% of all households including at least one person with a disability [19]. Families with disabilities were more likely to be in the lower socio-economic status groups, have larger household sizes and a higher dependency ratio with a lower proportion of household members who were working compared to households without a disabled family member [19].

The study was originally planned to take place in various areas within four of the 22 departments in Guatemala. However, due to mudslides during the autumn of 2017 and resultant road closures, interviews were conducted in only three regions: Guatemala City in Guatemala (urban); Tamahu, Coban in Alta Verapaz (semi-urban and rural); and Santiago Atitlan and Panajachel in Sololá (semi urban and rural). The areas were identified based on recommendations and partnership with local Organizations of Persons with Disabilities (OPD) working with CBM–Guatemala, a disability focused International Non-Governmental Organization with a long history of working in the area.

## Participant selection and data collection

Interview guides were developed by the primary author based on preliminary results of a meta-synthesis of qualitative studies, focusing on barriers to accessing primary healthcare services for people with disabilities [5] and in consultation with disability stakeholders in Guatemala and internationally. Interview questions inquired about individuals with disabilities' health conditions, both related to their impairments and general health conditions they may experience, how they differentiate between their general and specialized healthcare needs, how healthcare decisions are made in their household/family, their experience with accessing healthcare services to meet their healthcare needs and their suggestions on how access to primary healthcare can be improved. The guides were translated into Spanish by a registered Guatemalan interpreter to ensure cultural appropriateness. This process involved two teleconferences to ensure that the translated guides would match the original and minimize content lost in translation.

Participants with disabilities were identified through referrals from contacts by the local OPDs in the three different regions. A purposive sampling strategy was used, to try to include as broad a sample as possible, focusing on diversity in age, gender, and types of impairment. The inclusion criteria were adults over the age of 18 years with any impairments who could engage either independently or with the support of a caregiver or an interpreter in the interview. The only exclusion criteria were if the person was not able to give consent to participate in the study.

Interviews were conducted during the two separate periods in October 2017 and March 2018. Interviews during the initial data collection were completed by the primary investigator (GH) with support of a Spanish interpreter and research assistant (ALS). The primary investigator is a female occupational therapist trained in Canada and a PhD candidate in public health with extensive travel, clinical, and qualitative research experience in other LMICs. ALS completed the interviews during the second data collection period. She is a local Guatemalan from Guatemala City. ALS has family ties with both Coban in Alta Verapaz and the Western regions of Guatemala, including Sololá, providing her with familiarity and increased insight into some of the local customs of the populations in those regions. This allowed ALS to provide not only language translation but also cultural interpretation as appropriate for the primary investigator. ALS was trained on the importance of confidentiality and direct translation of content as much as possible to minimize changes to the content of the interview or misinterpretation during the translation process as well as interview processes for qualitative research.

Interviews took place primarily in the participant's home except for nine interviews that were completed in private spaces at OPD offices. The interviews took between 45 to 90 minutes depending on the need for interpretations, clarifications, and how much the participants wanted to share beyond the interview guide. Only nine of the total interviews were conducted in Spanish with the rest requiring a second interpreter for either sign language or a local dialect to Spanish interpretation. When a participant was unable to communicate independently or requested the presence of his/her caregiver, caregivers were invited to join the interview process, but only if the primary participant consented to complete the interview with the additional support. All interviews were audio recorded. Field notes were also taken throughout the interviews to supplement the interview notes and recordings.

## Data analysis

Data was analyzed using thematic analysis [20]. Interviews were transcribed by the primary interviewers into word documents and then anonymized and uploaded into NVivo 12 software for both data management and initial coding by the primary author. In addition to the coding performed by GH, six randomly selected transcripts were shared with MW, who also coded them independently. This process allowed for further validation of the inductive coding conducted by GH of those transcripts and MW to become somewhat familiar with the content of the interviews for additional opportunity for GH and MW to discuss the content and the categorization into themes.

Initial inductive coding of the interviews line by line resulted in a total of 31 original themes with several subthemes. The primary author then combined some of the themes following discussions with co-authors MW and HK. After various rounds of re-organizing the original themes and considering the various frameworks on access to healthcare and healthcare decision making, the primary author arrived at four primary themes, focusing specifically on the additional difficulties facing people with disabilities rather than difficulties seeking healthcare in general. The four primary themes: Perceived severity of illness and need for treatment, Personal attributes, Societal factors, and Health system factors, represent the variables that influence the decision to seek healthcare or not at the primary healthcare level, in addressing general healthcare conditions by people with disabilities. While categorized as individual themes, it is noteworthy that the themes may inter-relate and influence one another.

## Results

Twenty seven of 32 participants and caregivers who were approached by the research team completed the interviews. Sixteen Interviews were completed by the primary author (GH) during the initial data collection period and 11 were completed by ALS during the second data collection period. Three potential participants declined to participate during the second data collection period after finding out that they would not be provided with any compensation and two were excluded as they were unable to participate independently or consent to caregiver support to complete the interview.

Nineteen of the interviews took place in the Lago Atitlan region of Sololá due to the presence of the main partner OPD named ADISA, located in Santiago Atitlan. This region is a relatively popular area for both local and international travelers, with good ease of travel in the region in comparison to the other regions that were impacted by rainfall and mudslides during both periods of interviews in 2017 and 2018.

Fourteen men and 13 women participated in the interviews with the age ranging from 18 to 70 years of age (Table 1). The types of impairments were grouped into five categories based on the participants' reported primary difficulty and observed functional status: 1) mobility and

**Table 1. Distribution of participants based on their gender, age, and impairment.**

|  | Age group in years | | | | | | Type of impairment | | | |
|---|---|---|---|---|---|---|---|---|---|---|
|  | 18–25 | 26–35 | 36–45 | 46–55 | 56–65 | 66+ | Mobility | Psychosocial and cognitive | Sensory | Multiple |
| Men | 3 | 3 | 2 | 3 | 2 | 1 | 8 | 2 | 1 | 1 |
| Women | 3 | 3 | 0 | 0 | 5 | 2 | 8 | 4 | 3 | 0 |
| Total | 6 | 6 | 2 | 3 | 7 | 3 | 16 | 6 | 4 | 1 |

movement related difficulties due to neurological or physical impairments; 2) sensory impairments related to vision and/or hearing loss 3) psychosocial or cognitive impairments related to mental health or developmental status, and 4) multiple impairments.

None of the 27 participants lived on their own. Ten participants lived in households of three to five people with the rest living in households greater than six people (largest having 17 members). Seventeen participants reported being the only person in the household with a disability, 13 were married, and 19 had some level of schooling. Only five participants had access to some form of health insurance, private and/or social security. Nine of the participants were employed, seven were self-employed, one was retired and 10 were unemployed (Table 2).

When asked about what happened last time they had a general healthcare need, 15 participants stated that they sought some form of primary healthcare and 12 stated that they did not seek services and instead chose either to do nothing, use a home remedy, or wait. Cost was reported to be one of the primary factors affecting whether to seek primary healthcare by most participants. However, it quickly became apparent that while finances played an important role, there was a more complex internal process that influenced the decision about whether to seek primary health care services or not. The four primary themes identified demonstrate this complex decision-making process: 1) Perceived Severity of illness and need for treatment, 2) Personal attributes, 3) Societal factors, and 4) Health system characteristics.

## 1. Perceived severity of illness and need for treatment

This theme refers to the individual with a disability or their families' perception of how sick they are, their perception of their need for healthcare services at the time, and the perceived benefits of making a healthcare visit in their presenting condition. It expands on Kleinman's beliefs component of the 'popular sector' focusing on the beliefs a person holds about their illness and the meaning attached to their condition [16]. It also includes their expectations regarding how a healthcare provider may treat them and what the outcome of the interactions may be [16,21].

The following quote demonstrates an example of these perceptions as the 20-year-old woman with psychosocial impairment explains how her mother has convinced her that there is no value to going to the health center, given her presenting conditions, due to the time and effort involved: '*I have not gone to the health centre or the doctor, not only because of lack of answer . . .. Sometimes. We think, why am I going to go, spend money to go to the health centre*

**Table 2. Distribution of participants based on social factors of household size, marital status, access to insurance, employment status and level of education.**

|  | Household size (number of people) | | | Marital status | Some education | Health insurance | | | Employment status | | | |
|---|---|---|---|---|---|---|---|---|---|---|---|---|
|  | 3–5 | 6–10 | 11+ | Married |  | None | private | public | Employed | unemployed | Self employed | retired |
| Men | 5 | 7 | 2 | 8 | 12 | 14 | 0 | 0 | 6 | 4 | 4 | 0 |
| Women | 5 | 4 | 4 | 5 | 7 | 8 | 2 | 3 | 3 | 6 | 3 | 1 |
| Total | 10 | 11 | 6 | 13 | 19 | 22 | 2 | 3 | 9 | 10 | 7 | 1 |

*or to go to the doctor and they are going to do nothing, so my mom said, don't go out. You are not going to go out from now on.'*

In the next quote, a 56- year-old woman with hearing impairment explains that she prefers to wait for her condition to worsen then seek emergency care, stating that her condition will be taken more seriously if she is sicker and, in an emergency, setting and as a result worth the effort: *'I really have to think about it. When I feel very sick then I decide to go to the emergency. If I go to a regular service, I have to wait a lot but in emergency I am seen faster. I prefer to go to emergency and don't self-medicate, because if I don't know what is going on maybe I will die. . .. When I was feeling sick to my stomach. . .. [and went to the health centre . . . . . .It is very difficult. It doesn't matter if you are feeling very bad. . ..so easier to go to emergency, it is faster and all in one place.'*

While perceived severity of an illness is clearly part of the decision-making process on whether to seek care at the primary healthcare center or not, it is only one of multiple factors taken into consideration. It is also important to note that the lived experience of having a disability related to nature of the impairment, whether congenital, acquired, traumatic, or progressive, and duration living with disability can also influence a person's perception of severity of their illness and need for treatment.

## 2. Personal attributes

Personal attributes represent factors that are unique to a person and focuses on the essence of the individual, in this case a person with a disability. In addition to demographics, these attributes may include personal experiences, characteristics such as traits and abilities, and values and goals, despite what one may choose to do based on cultural and behavioural expectation.

While personal attributes are often stable, some may change over time such as acquiring a disability and how the experience may influence or shape one's abilities, values, and goals. This theme builds on the 'Biology and Genetic Predisposition' component of the Health Capability Model but adds an additional layer related to personality characteristics and traits borrowed from Kleinman's model related to how one perceives the significance and impact of an illness or impairment. For example, some elements of personal attributes, like ability to stand for long periods of time are very physiological and clear, as indicated by the following quote from a 70-year-old woman with mobility impairments: *'They have long lines of people waiting and I can't wait for long time because of my legs'.* Other attributes, however, may be less concrete and are more about values, personality traits, or previous experiences of an individual as exemplified in the following quote from a 29-year-old woman with hearing impairment: *'When I think about going to the doctor, I first think I have to go with my mom. I never go alone. I am scared I will make a mistake or about the schedule'* and a 24-year-old woman with mobility impairment: *'The thing is that sometimes I have had difficulty going to a healthcare service because of the way people treat you to have access. . .. So I have had problems because of the transportation. . .. It is very hard, but I have to do it. I have no choice. I am always going by myself. No one helps me'.* In the first quote the participant was born with a hearing impairment and feels scared to access services on her own while in the second quote the participant, who has only lived with her disability for two years, recognizes that despite the hardship, she has "no choice" but to prioritize her wellbeing and goes on her own.

## 3. Societal factors

Societal factors represent social and contextual components external to an individual and their connectedness to this external system. Societal factors are best described by combining the social sector and community in the 'popular sector' of the Explanatory Model of Illness [16] and combining it with the 'macro, social, political, and economic environment' and portions

of the 'intermediate social context' of the Capability Model [15]. This means that the theme contains a substantial number of factors dictated by the socio-economic and political forces and pressures, also known as structural factors, which influence individual choices and social interactions and expectations related to cultural values and religious beliefs about health and illness. While societal values and beliefs may or may not actually be in line with the values and attitudes of the person with a disability (their personal perspectives), they do influence the negotiation process that is involved for the disabled person and their decisions about what they feel they can or should do as opposed to what they may wish to do with respect to seeking healthcare. For example, the following quote demonstrates how a 21-year-old woman with psychosocial impairments is influenced by her perception of social beliefs around people with disabilities and their treatment at the health center: '*No, I have not visited any doctors. Because we don't receive any attention in that place. I remember once they took a blood sample and then told me I don't have anything (pointing to her fingertip- indicative of blood sugar). I don't remember when it was. . .. I felt that, I had the idea that I was treated differently because of the family. I heard that they said that I am just lying, I am just saying that I feel bad, but I don't have anything. And I just came back without being checked or being treated*'. While it is not clear whether the participant's perceptions are accurate, they are real to them and influence her actions and decision making.

The next quote by a 54-year-old-man with mobility impairment demonstrates how despite his desire to seek healthcare for infections related to burn wounds, he does not seek services at primary care facilities (the drug store or the health center) due to competing priorities and relies on his cousin who is a nurse:

'*. . .the inconvenience is that the health center doesn't have the resources, instruments, or medicines to receive treatment [for infections on his skin]. Finances are the biggest problem. we need money to cover expenses of the family, utilities at home. My cousin, he is a nurse. When he has vacations he comes to the house. I am trying to do my best to feel better and to be optimistic about my condition. I hope to recover totally, but understand I am older and may not recover well, so am satisfied to know that I still have some days alive and that's ok. I am grateful with the "extra" days I have.*'

The above example also demonstrates the value of connectedness and access to resources beyond the formal system of care. Finally, the next quote by the wife of a 50-year-old man with mobility impairment emphasizes the multi-dimensional nature of decision making related to the structures and systems, including physical and attitudinal components of the society when use of primary healthcare services is considered by a disabled person: '*He hasn't visited the medical center ever. The reason is because they don't pay attention. Once he visited the CAIM (health center). There was a psychologist and she didn't say anything. She just was asking about what happened and didn't transfer him to the doctor. Additionally, it becomes more expensive to go because he can't go by himself, he has to go with someone, and it is not affordable*'.

This theme reveals how the external context and environment that shapes people's experiences of health and illness, is complicated by the presence of an impairment. It can result in inaccurate assumptions, miscommunications, and discrepancies in expectations associated with accessing primary healthcare services for people with disabilities.

## 4. Health system characteristics

Health system characteristics are the elements related to the provision of healthcare services and resources including medicine, medical equipment, location, accessibility of health

facilities, number and types of healthcare providers and their training. Similar to other themes, this also intersects with parts of the Explanatory Model of Illness and the Health Capability Model, particularly the 'Professional sector' and the 'public health and healthcare system' respectively [15,16]. This theme covers a range of health system characteristics that impact desirability or access to services for people with disabilities. While some issues, such as affordability and wait times, may be experienced by many people, they are enhanced for people with disabilities because of their on average higher levels of poverty and pre-existing health condition related to their impairment. Other issues, such as accessibility and negative attitudes however may be specific to people with disabilities as described by the following quote from a 43-year-old man with a mobility impairment: '*The problem with it is that it is not very accessible and sometimes people or professionals are not very respectful at the health clinic. . . ..The first thing is that I have seen they do not respect people with disability. In my case, I don't expect to be given priority but in other cases I have seen people need priority and they professionals do not take that into consideration. Also, they are not aware of disabilities*'.

## Discussion

While it is important to note that many people with disabilities do make the decision to access and use primary healthcare services when they feel the need, the decision-making process undertaken by people with disabilities about whether to seek primary healthcare services or not is extremely complex and is influenced by several variables that interact in ways that can be unique for everyone. As the results demonstrate, the decision to access primary healthcare services for this population often occurs in the context of the family, within a household with unique circumstances, influenced by the above four themes and their interactions with one another. This is similar to what Kleinman (1978) describes as a "psychocultural network" of beliefs and experiences with a cognitive process that occur within a "pluralistic system", consisting of individuals, in this case disabled people, their family, and the healthcare system. While there are similarities between the general population and people with disabilities in some of the barriers they face in accessing healthcare, the results show that due to their often-marginalized position in society, people with disabilities may not only experience the common barriers more strongly, but also face unique barriers related to or specific to their impairments and resultant disability.

As previously discussed, research has shown that people with disabilities' choice to seek healthcare services or not, is influenced by three types of barriers: cultural beliefs or attitudinal barriers, informational barriers, and practical or logistical barriers [5]. While these barriers can occur at the various entry points within the healthcare seeking journey it is anticipated that they may be particularly influential during the decision-making process that starts at home. For instance, informational barriers can influence the decision-making process in terms of perceived severity of illness and need for treatment in several ways. Given someone's personal attributes including their experience of disability, they may either minimize their symptoms and relate it to the underlying condition related to their impairment or they may become more anxious thinking their underlying condition is worsening or alternatively feel that it is not even related to their underlying condition (which may be the case), all dependent on the knowledge and information they have related to their specific conditions or other aspects of health.

Given this complexity, the following conceptual model of healthcare decision-making for people with disability is offered to illustrate the inter-related and dynamic nature of the decision-making process that occurs when trying to decide if they should seek primary healthcare services (Fig 4). The model draws on the themes that emerged both from this study and the

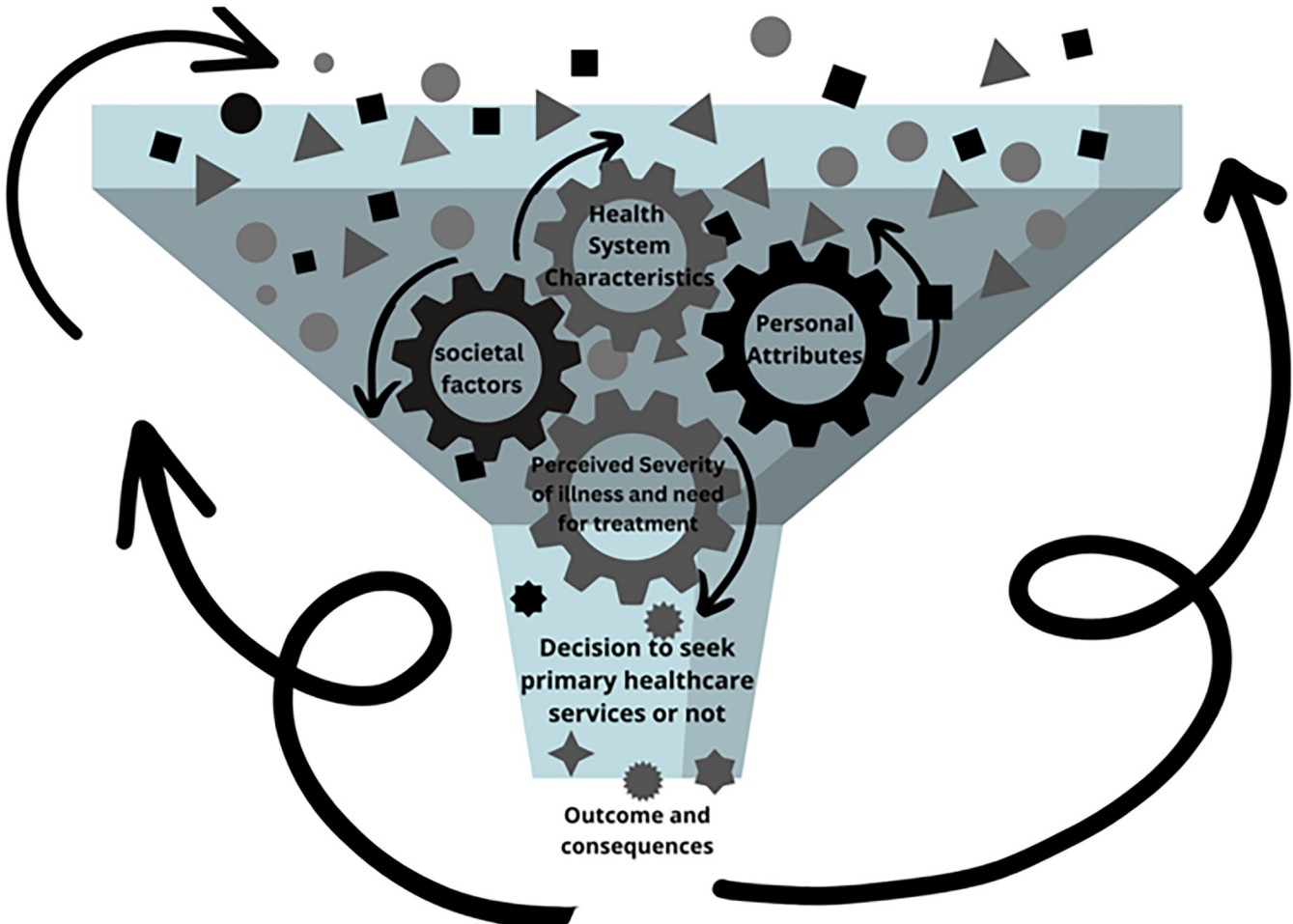

**Fig 4. Conceptual model for decision making on utilizing primary healthcare services by people with disability.**

meta-synthesis on barriers to accessing primary healthcare services for people with disabilities in LMIC [5].

In this model, the funnel represents the person with a disability with a health condition and all the various factors that interact and influence their decision-making process on accessing primary health care services at the household level. The cogwheels represent the four themes identified and how their rotation and size, can influence the decision-making process. The cogs in the cogwheel represent the multidirectional influence of the themes and how they can overlap or influence one another as well as be influenced by outcomes from past decisions and experiences and the barriers that exist. The model implies that while separate themes have been described, none of the components in the framework can be isolated, even if their influence is diminished or increased based on the ongoing and changing circumstances of personal attributes, societal factors, health system characteristics and the perceived severity of illness and need for treatment.

What this model aims to demonstrate is that in addition to addressing obvious barriers to accessing primary healthcare, once the health seeking journey has begun, interventions should also target households, where the transaction between the person, in this case one who has a disability, and their family occurs. The members of a household each contribute their own perspectives based on their own experiences and beliefs about impairment, disability, and illness, influenced by their own experiences. This multi-person transaction can result in discrepancies in expectations and increased barriers in accessing primary healthcare services due to informational, cultural and attitudinal, and practical and logistical barriers. A household approach, however, cannot be implemented in isolation. The success of such an intervention will also require a community-based approach that will also address societal factors and health system characteristics and hence the overall accessibility and quality of the service for people with disabilities.

### Limitations of the study

Despite every effort to ensure diversity in age, gender, disability, and ensuring cultural understanding of the participants in this study, it is important to note that the study has several limitations that may have impacted the results.

One set of limitations is related to the need for interpreters and the presence of a non-Guatemalan interviewer during the first data collection period. It is possible that due to the need for interpreters and at times two interpreters, some components of the responses and perhaps key details may have been missed or lost in translations. In addition, during the second round of data collection when the foreign investigator was not present, the loss of potential participants may have been due to the individuals feeling less inclined to participate or more comfortable with declining an interview from a fellow Guatemalan, indicating that some responses may have been influenced by the presence of a foreign investigator.

Another limitation is related to the geographical locations covered. This was influenced by both the presence of an OPD that was able to refer participants and accessibility between the investigator and potential participants in areas impacted by flooding and mudslides. This highlights the increased barriers faced by people with disabilities, excluding individuals who may have lived more remotely or have even less access to resources for transportation.

Finally, of note is the over representation of participants with visible disabilities, particularly physical or mobility impairments. This may be related to communities' understanding and definition of disability and the ongoing stigma related to mental illness or cognitive impairments.

### Conclusion

This study contributes to the work on access and utilization of primary healthcare services for people with disabilities in Guatemala through proposing an innovative and new conceptual model that focus on the specific circumstances faced by people with disabilities. The model demonstrates the complexity of the decision-making process involved in seeking primary healthcare services for people with disabilities in LMIC and can facilitate the development of more effective interventions and policies to improve access to primary healthcare services for people with disabilities. Improving access to healthcare services, starting with primary healthcare for disabled people should focus not only on the experience of healthcare but the entire health seeking journey, including how healthcare decisions are made by people with disabilities and their household, starting at home.

### Supporting information

**S1 Checklist. Inclusivity in global research.**
(DOCX)

## Acknowledgments

We would like to thank all participants and their caregivers who participated in this study for their time and insights. We would also like to thank CBM-Guatemala, ADISA, and their partner organizations for referring us to potential participants and supporting us in the provision of sign and local language translations.

## Author Contributions

**Conceptualization:** Goli Hashemi, Mary Wickenden.

**Data curation:** Goli Hashemi, Ana Leticia Santos.

**Formal analysis:** Goli Hashemi.

**Funding acquisition:** Hannah Kuper.

**Investigation:** Goli Hashemi.

**Methodology:** Mary Wickenden.

**Project administration:** Goli Hashemi.

**Supervision:** Mary Wickenden, Hannah Kuper.

**Validation:** Mary Wickenden.

**Writing – original draft:** Goli Hashemi.

**Writing – review & editing:** Goli Hashemi, Mary Wickenden, Hannah Kuper.

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
