## [Decision Letter · Decision Letter 0]

25 Apr 2022

PGPH-D-22-00569

How do people with disabilities in Guatemala make healthcare decisions? a qualitative study focusing on use of primary healthcare services

Dear Dr. Hashemi,

Thank you for submitting your manuscript to PLOS Global Public Health. After careful consideration, we feel that it has merit but does not fully meet PLOS Global Public Health’s publication criteria as it currently stands. Therefore, we invite you to submit a revised version of the manuscript that addresses the points raised during the review process.

Please submit your revised manuscript by . If you will need more time than this to complete your revisions, please reply to this message or contact the journal office at globalpubhealth@plos.org. Please include the following items when submitting your revised manuscript:

We look forward to receiving your revised manuscript.

Kind regards,

Manish Barman, MD., MSc., FRCP

Academic Editor

Journal Requirements:

1. Please insert an Ethics Statement at the beginning of your Methods section, under a subheading 'Ethics Statement'. It must include:

a) The name(s) of the Institutional Review Board(s) or Ethics Committee(s)

b) The approval number(s), or a statement that approval was granted by the named board(s) 

2. Please include a complete copy of PLOS’ questionnaire on inclusivity in global research in your revised manuscript. Our policy for research in this area aims to improve transparency in the reporting of research performed outside of researchers’ own country or community. The policy applies to researchers who have travelled to a different country to conduct research, research with Indigenous populations or their lands, and research on cultural artefacts. The questionnaire can also be requested at the journal’s discretion for any other submissions, even if these conditions are not met.  Please find more information on the policy and a link to download a blank copy of the questionnaire here: https://journals.plos.org/globalpublichealth/s/best-practices-in-research-reporting. Please upload a completed version of your questionnaire as Supporting Information when you resubmit your manuscript.

3. Please upload all main figures as separate Figure files in .tif or .eps format only and removed from manuscript file. For more information about how to convert and format your figure files please see our guidelines:

4. Since your data is not available for proprietary reasons, please explain via email why the data is not available. Please also include the contact information for the third party organization that should be contacted should other researchers want to request access to this data and please include the full citation of where the data can be found. We also request that you verify with us via email that any researcher will be able to obtain the data set in the same manner that the you have obtained it. If you feel you are unwilling or unable to adhere to this policy, please explain your reasons by return email and your exemption request will be escalated to the editor for approval. Your exemption request will be handled independently and will not hold up the peer review process, but will need to be resolved should your manuscript be accepted for publication. One of the Editorial team will be in touch if they require more information.

Additional Editor Comments (if provided):

Dear Authors

Excellent article overall

You should try incorporate some minor changes as suggested by the reviewers.

If possible try a professional editor for easy readability and flow of words.

Looking forward to your revision.

Regards

Reviewers' comments:

Reviewer's Responses to Questions

**Comments to the Author**

1. Does this manuscript meet PLOS Global Public Health’s publication criteria? Is the manuscript technically sound, and do the data support the conclusions? The manuscript must describe methodologically and ethically rigorous research with conclusions that are appropriately drawn based on the data presented.

Reviewer #1: Yes

Reviewer #2: Yes

2. Has the statistical analysis been performed appropriately and rigorously?

Reviewer #1: Yes

Reviewer #2: Yes

3. Have the authors made all data underlying the findings in their manuscript fully available (please refer to the Data Availability Statement at the start of the manuscript PDF file)?

Reviewer #1: Yes

Reviewer #2: Yes

4. Is the manuscript presented in an intelligible fashion and written in standard English?

Reviewer #1: Yes

Reviewer #2: Yes

5. Review Comments to the Author

Reviewer #1: 1) Overall, this study identified a clear research gap and generated four major themes which addresses the gap in the decision making process and its key determinants for accessing primary health services by persons living with disability (PWD) in 3 regions of Guatemala.

2) Guatemala has 22 departments across 8 regions. This qualitative survey was conducted in only 3 regions and as such the findings cannot be generalized for the entire country. The country has 10% of its 18 million population categorized as disabled with 1 in 3 households having a person living with disability. Therefore, interviewing 27 PWD cannot reflect the entirety of Guatemala. I suggest that the title and methods should reflect ‘’….in three regions of Guatemala’’ and not just Guatemala.

3) Put lines 287-292 in italics to reflect the quote and same for lines 311-312, 315-316 and 317-319

Reviewer #2: Comments:

Dear authors your manuscript is well designed and well drafted I appreciate it.

I have only two comments on your manuscript.

Add citation in your discussion part by referring the previous articles to make your study more strong.

Figure 1 and 2 are not visible, try to make it visible.

In general this manuscript needs minor revision.

6. PLOS authors have the option to publish the peer review history of their article (what does this mean?). If published, this will include your full peer review and any attached files.

**Do you want your identity to be public for this peer review?** For information about this choice, including consent withdrawal, please see our Privacy Policy.

Reviewer #1: No

Reviewer #2: No

---

## [Editor Report · Decision Letter 1]

22 Sep 2022

PGPH-D-22-00569R1

How do people with disabilities in three regions of Guatemala make healthcare decisions? a qualitative study focusing on use of primary healthcare services

Dear Dr. Hashemi,

Thank you for submitting your manuscript to PLOS Global Public Health. After careful consideration, we feel that it has merit but does not fully meet PLOS Global Public Health’s publication criteria as it currently stands. Therefore, we invite you to submit a revised version of the manuscript that addresses the points raised during the review process.

We look forward to receiving your revised manuscript.

Kind regards,

Manish Barman, MD., MSc., FRCP

Academic Editor

Journal Requirements:

2. Please include a complete copy of PLOS’ questionnaire on inclusivity in global research in your revised manuscript. Our policy for research in this area aims to improve transparency in the reporting of research performed outside of researchers’ own country or community. The policy applies to researchers who have travelled to a different country to conduct research, research with Indigenous populations or their lands, and research on cultural artefacts. The questionnaire can also be requested at the journal’s discretion for any other submissions, even if these conditions are not met.  Please find more information on the policy and a link to download a blank copy of the questionnaire here: https://journals.plos.org/globalpublichealth/s/best-practices-in-research-reporting. Please upload a completed version of your questionnaire as Supporting Information when you resubmit your manuscript.

3. Please amend your detailed Financial Disclosure statement. This is published with the article. It must therefore be completed in full sentences and contain the exact wording you wish to be published.

4. We have noticed that you have uploaded Supporting Information files, but you have not included a list of legends. Please add a full list of legends for your Supporting Information files after the references list. 

Additional Editor Comments (if provided):

Dear Authors

The revision is satisfactory except-

The Figures now added as separate tiff files are still quite unclear and hazy.

It seems the Figures are not original by the authors and are probably sourced from somewhere.

This might have copyright issues both for you and the journal.

Either these figures should be properly referenced after taking permission from the original sources or an alternative way can be if Authors can draw /create there own figures with some modifications.

Looking forward to your re submission ASAP

Thanks

Editor
---

## [Editor Report · Decision Letter 2]

21 Dec 2022

PGPH-D-22-00569R2

How do people with disabilities in three regions of Guatemala make healthcare decisions? a qualitative study focusing on use of primary healthcare services

Dear Ms. Goli Hashemi,

Thank you for submitting your manuscript to PLOS Global Public Health. After careful consideration, we feel that it has merit but does not fully meet PLOS Global Public Health’s publication criteria as it currently stands. Therefore, we invite you to submit a revised version of the manuscript that addresses the points raised during the review process.

In the data analysis section, Lines 222-224, the authors wrote: "The results were then shared with MW, who reviewed 20% of the interviews for interrater coding and discussion as necessary to ensure agreement and consensus." 

This sentence is highly charged and unclear to me vis-a-vis the trustworthiness that is expected when handling qualitative data. The phrase "reviewed 20% of the interviews for interrater coding" was most confusing and I struggled to understand what the authors did practically and what the implications are regarding the trustworthiness of the study. I also wondered if having the knowledge of 20% of the interview content can lead to a meaningful discussion "to ensure agreement and consensus.

We look forward to receiving your revised manuscript.

Kind regards,

Ferdinand Mukumbang, PhD

Academic Editor

Journal Requirements:

b. If any authors received a salary from any of your funders, please state which authors and which funders.
---

## [Editor Report · Decision Letter 3]

13 Jan 2023

How do people with disabilities in three regions of Guatemala make healthcare decisions? a qualitative study focusing on use of primary healthcare services

PGPH-D-22-00569R3

Dear Ms. Goli Hashemi,

We are pleased to inform you that your manuscript 'How do people with disabilities in three regions of Guatemala make healthcare decisions? a qualitative study focusing on use of primary healthcare services' has been provisionally accepted for publication in PLOS Global Public Health.

Best regards,

Ferdinand Mukumbang, PhD

Academic Editor

Please, review the grammar. Data is plural not singular.